# HRV-Guided Training for Elders after Stroke: A Protocol for a Cluster-Randomized Controlled Trial

**DOI:** 10.3390/ijerph191710868

**Published:** 2022-08-31

**Authors:** María Carrasco-Poyatos, Antonio Granero-Gallegos, Ginés D. López-García, Rut López-Osca

**Affiliations:** 1Department of Education, Health and Public Administration Research Center, University of Almeria, 04120 Almeria, Spain; 2Department of Education, University of Almeria, 04120 Almeria, Spain

**Keywords:** heart rate variability, HIIT, cardiac rehabilitation, safety, adherence, VO_2peak_, protocol

## Abstract

There are many consequences associated with having a stroke, all of which are important factors affecting long-term rehabilitation outcomes; these become important health issues for those of advanced age and require dedicated health strategies. High-intensity interval training (HIIT) is an effective training protocol used in cardiac rehabilitation programs; however, owing to the inter-individual variability in physiological responses to training associated with cardiovascular diseases, the exercise regimen given to each patient should be closely controlled and individualized to ensure the safety and efficiency of the exercise program. Heart rate variability (HRV) is currently being used for this purpose, as it is closely linked to parasympathetic nervous system activation, with high HRV scores associated with good cardiovascular adaptation. The objective of this protocol is to determine the effect of HIIT compared to HRV-guided training in terms of cardiorespiratory fitness, heart rate variability, functional parameters, body composition, quality of life, inflammatory markers, and cognitive function in patients who have had a stroke, as well as to assess the feasibility of patients undertaking an 8-week cardiac rehabilitation program, evaluating its safety and their adherence. The proposed protocol involved cluster-randomized controlled design in which the post-stroke patients are assigned either to an HRV-based training group (HRV-G) or a HIIT-based training group (HIIT-G). HIIT-G will train according to a predefined training program, whereas HRV-G will train depending on the patients’ daily HRV. The outcomes considered are peak oxygen uptake (VO_2peak_), endothelial and work parameters, heart rate variability, functional parameters, relative weight and body fat distribution, quality of life, inflammatory markers and cognitive function, as well as exercise adherence, feasibility, and safety. It is expected that this HRV-guided training protocol will improve functional performance in patients following a stroke and be safer, more feasible, and generate improved adherence relative to HIIT, providing an improved strategy for to optimize cardiac rehabilitation interventions.

## 1. Introduction

Non-communicable chronic diseases are among the main health problems worldwide. According to the U.S. Department of Health and Human Services, cardiovascular diseases, diabetes mellitus, and obesity account for approximately two-thirds of deaths globally and are responsible for 73.5% of total deaths [1]. Individuals tend to accumulate coexisting diseases as they age. With advanced age, patients with risk factors for atherosclerosis would be expected to show multiple cardiac, aortic, extracranial, intracranial, and penetrating artery occlusive lesions that pose a stroke risk [2]. Therefore, it is not surprising that 88% of subjects with a first diagnosis of chronic heart disease are aged between 65 and 80 years [3], making this a common pathology among older people.

A stroke is a cardiovascular disease that affects the blood vessels supplying blood to the brain [4], resulting in nerve cell death in the affected brain tissue [5]; therefore, strokes are also known as brain infarcts, a heterogeneous disease with more than 150 known causes that can be categorized as ischemic or hemorrhagic strokes, subarachnoid hemorrhages, cerebral venous thrombosis, or spinal cord strokes [6]. The pathophysiological processes involved after a stroke include bioenergetic failure, acidosis, loss of cell homeostasis, excitotoxicity, and disruption of the blood–brain barrier [5], generating an inflammatory process. According to Pecherstorfer et al. (2009) [4], the consequences of a stroke are manifold, depending on the location of the affected brain regions, and can involve physical, cognitive, emotional, sexual, and vocational impairment, as well as social handicap. All of these are important factors affecting long-term rehabilitation outcomes and become important health issues for those of advanced age, requiring dedicated health strategies.

One such strategy is physical exercise, which has become an important tool in cardiac rehabilitation programs. Taylor et al. (2004) [7], in a systematic review with meta-analysis, pointed out that the physical training included in cardiac rehabilitation programs is associated with a reduction of 26% in the patient mortality rate. More recently, in another systematic review with meta-analysis, Kraus et al. (2019) [8] determined that low volumes of exercise were enough to produce health benefits in cardiovascular disease patients and that vigorous physical exercise reduced the mortality risk by 75% in this population. In this regard, it is well established that high-intensity interval training (HIIT) is an efficient and safe training protocol for improving VO_2_ [9] and other physical and physiological parameters [10], reducing the risk of cardiovascular diseases.

On the other hand, the anti-inflammatory response after HIIT has been investigated in middle-aged and older people, with no consensus reached between studies. According to the two systematic reviews with meta-analysis mentioned above, resistance training has a positive effect on anti-inflammatory biomarkers [11], whereas the effect of HIIT is still unclear [12]. Moreover, it is essential to evaluate hippocampal functionality in order to determine the cognitive effects of improved cardiac functions and improved cognition associated with the practice of physical exercise with aging [13]. Therefore, it would be interesting to determine whether different training regimes generate different cognitive adaptations following a stroke.

However, cardiac rehabilitation is often carried out in groups, and it is recognized that group training using the same standardized training program can result in a wide range of reactions in terms of performance and physiological adaptations [14,15]. Given that an individual’s cardiovascular system can be affected by a stroke to varying degrees, the exercise regime given to each patient should be closely controlled and individualized, thus avoiding overstimulation of the cardiovascular system during training while increasing safety and efficiency. However, serious adverse events still occur in cardiac rehabilitation during HIIT or moderate-intensity training [16]. As stated by Schmitt et al. (2018) [17], an important component of the interindividual variability in the physiological responses to training is related to the balance between the parasympathetic (PNS) and sympathetic (SNS) activity of the autonomic nervous system (ANS). In this regard, heart rate variability (HRV) is currently used as an indicator of the physical and psychological state of athletes, enabling coaches and scientists to individualize training.

HRV is defined as the fluctuation in the interval between consecutive heartbeats and the fluctuation between consecutive instantaneous heart rates [18]. Therefore, it is linked to the functioning of the autonomic nervous system and the fluctuation between sympathetic and parasympathetic activation. Healthy subjects are usually characterized by a predominantly parasympathetic system at rest, involving a low heart rate and high HRV and demonstrating improved cardiovascular system adaptation and a positive state of health [19]. In contrast, cardiovascular diseases, such as hypertension, ventricular arrhythmia, chronic heart failure, or ischemic heart disease, increase the predominance of the sympathetic system, which is reflected in a high resting heart rate and low HRV [20,21,22]. According to the New York Heart Association, there is a strong correlation between low scores in HRV temporal parameters and mortality after a stroke [23]. Moreover, Daminello et al. (2013) [24] concluded that in stroke patients, during exercise, HRV decreases and does not return to baseline within the first 30 min after exercise. Thus, they considered HRV an important method for appropriate prescription of exercise for this group of patients.

HRV can be used not only to detect cardiovascular disease severity but also to reflect a reduction in disease impact. This means that HRV can be used in any therapy that aims to shift the autonomic equilibrium toward parasympathetic dominance and a reduction in the sympathetic tone [25], with the objective of increasing homeostasis of the autonomic nervous system. To this end, training should focus on cardiovascular system adaptability. In this regard, HRV biofeedback has achieved positive results by maximizing the heart rate increment during inspiration and reducing it during exhalation by means of various breathing techniques [26,27]. Similarly, Nishime et al. (2000) [28] reported that recovery after an exercise test serves as a good indicator of cardiac disease rehabilitation, as better recovery indicates a lower mortality rate. According to López-Chicharro et al. (2013) [29] and Hackney et al. (2006) [30], training at moderate-to-high intensity is related to a reduction in sympathetic activation. Moreover, a recent systematic review with meta-analysis revealed that basal HRV can increase following HIIT training [31].

Therefore, HRV is a biomarker that can indicate the health status of the cardiovascular system, serving as an effective tool for individualizing the training load in cardiac patients. Moreover, it can be used to measure cardiovascular responses to therapeutic interventions, with HIIT-based exercise being the most recommended option. However, with the goal of obtaining improved results regarding cardiovascular rehabilitation, it is still unclear whether to opt for HIIT training or to individualize HIIT training using HRV. We hypothesize that (i) both HIIT training and HRV-guided training will improve cardiorespiratory fitness, functional performance, heart rate variability, relative weight and fat distribution, inflammatory markers, cognitive function, and quality of life in post-stroke patients but that HRV-guided training will achieve better results; and (ii) HRV-guided training will be safer and more feasible and will generate more patient adherence to the program. The primary objective of the proposed protocol is to determine the effect of HIIT- compared to HRV-guided training on cardiorespiratory fitness in post-stroke patients who undertake an 8-week cardiac rehabilitation program. The secondary aims of this trial are (i) to determine the effect of HIIT training compared to HRV-guided training on heart rate variability scores in post-stroke patients, (ii) to investigate the effect of each exercise program in terms of the adaptations obtained in functional parameters, body composition, and the quality of life of these patients, (iii) to study the effect of both programs on inflammatory markers, (iv) to determine which exercise program is more effective at improving cognitive function, and (v) to analyze whether the two exercise programs are equally feasible and safe, as well as to evaluate their effect on patient adherence to the cardiac rehabilitation program.

This will be an 8-week, single-center, prospective, cluster-randomized controlled trial protocol in which post-stroke patients are assigned to either an HRV-based training group (HRV-G) or a HIIT training group (HIIT-G). A block randomization method will be used to randomly assign groups to interventions in equally sized sample groups. This protocol was designed in accordance with the Standard Protocol Items: Recommendations for Interventional Trials (SPIRIT) Statement [32]. To describe the intervention, the TIDieR (Template for Intervention Description and Replication) checklist by Hoffmann et al. (2014) [33] was used.

## 2. Materials and Methods

### 2.1. Study Setting

To select a sample with the desired characteristics, University Hospital Torrecárdenas, (Almería, Spain) was chosen as the location to carry out this protocol. The room rate assigned to the cardiology section in this hospital is up to 0.77%, and re-admission as a result of cardiovascular diseases has increased by 64.79% in the past year. Within the internal organization system, two phases of care are considered for patients with cardiovascular events. In phase 1, the patient is attended by a cardiologist, who is responsible for detecting the origin of the event, sending the patient to surgery (if necessary) and prescribing medication. The physical condition level is also determined in phase 1 using a treadmill test (a modified Bruce protocol). Then, patients are invited to participate in phase 2, which consists of an 8-week cardiovascular rehabilitation program. This is conducted by a multidisciplinary group comprising a rehab doctor, a physiotherapist, and a nurse. The proposed protocol will be implemented in phase 2.

### 2.2. Eligibility Criteria

To be included in the program, participants must (i) have a left ventricle ejection fraction higher than 30% after stroke and (ii) be aged between 18 and 80 years. As exclusion criteria, the following will be taken into account: (i) the presence of absolute or relative contraindications for accomplishing the treadmill test, as indicated by the Spanish Society of Cardiology; (ii) ongoing treatment for other diseases or regularly taking a drug(s) that has/have a direct or indirect effect on the nervous system (e.g., anxiolytics, antidepressants, or neuroleptics); (iii) the absence of medication to control or modify cardiovascular disease during the intervention; (iv) people who are participating or have participated in other similar exercise programs in the previous three months; and (v) inability to perform at least 80% of the workouts during the intervention.

The cardiologist from University Hospital Torrecárdenas will verify that the patients meet the inclusion criteria. The trial steering members will certify that the subjects do not meet any of the exclusion criteria. The study requirements will be explained to patients in detail verbally and via the written patient information and consent form. The model consent form is shown in Appendix A.

### 2.3. Interventions

To implement this protocol, it will be necessary to include a physical exercise graduate in the multidisciplinary group (described in Section 2.1), who will design and conduct the intervention. This will ensure that patients are controlled, monitored, and directed by a specialist at all times, thus guaranteeing the safety of the protocol and optimizing the intervention.

The intervention program will be conducted over a period of 8 weeks, with three one-hour sessions/week. Every session will comprise a warmup, the main intervention, and a cooldown. The warmup and cooldown will last 10–12 min, respectively, and will be based on the dynamic mobilization of the main muscle groups, especially those of the lower body, to prepare and recover them, respectively, for the main part of the intervention. In the main part, lasting 30–35 min, aerobic-based exercise will be implemented. To reduce mechanical limitations while carrying out the exercise, both a treadmill and a cycloergometer will be used, rotating them weekly, as recommended by a recent meta-analysis [34]. Motivational strategies, such as playing music during the session and giving messages of encouragement, will be used by the professionals supervising the activity. The periodization of the intervention will include two training periods: a two-week familiarization period (FP) and a 6-week training period (TP). The FP will be common to both groups and will be used to increase the training intensity gradually to prepare the patients for the high-intensity TP. In the TP, each group will carry out the corresponding intervention. The HIIT-G group will train according to a predefined high-intensity training program. Taking into account that VO_2_ can be improved either with short or long HIIT intervals [9] and that patients are expected to be in poor physical condition, short training intervals of 1′30′′ will be combined with active rest periods. In each session, the work time at the determined intensity will be at least 50% of the total work interval. The training interval intensity will be gradually increased as the rest periods are reduced. In accordance with Taylor et al. (2017) [35], both the maximal heart rate and the subjective perception of effort will be controlled. The HIIT-G training periodization is presented in more detail in Table 1.

The training prescribed to the HRV-G group will depend on the subjects’ daily HRV and will follow the HRV-training schema designed by Kiviniemi et al. (2007) [36] (Figure 1). The HRV registration procedure and analysis are described by Carrasco-Poyatos et al. (2020) [37], although, in accordance with the results from recent randomized controlled trials [38,39] and the characteristics of the sample, only high or low intensity will be considered for the present protocol.

The assigned study intervention will be modified or discontinued if the drug dose changes in response to harms, the participant requests it, the disease worsens, or the cardiologist recommends it. Compliance with the exercise protocol will be controlled by the nurse.

### 2.4. Outcomes

The primary outcome of this study will be the peak oxygen uptake (VO_2peak_) obtained in an incremental treadmill test. The secondary outcomes will be treadmill endothelial and work parameters; heart rate variability; functional parameters; relative weight and body fat distribution; quality of life; inflammatory markers; cognitive function; and exercise adherence, feasibility, and safety. The treadmill test and blood sample collection will be carried out by the cardiologist one week before and after the intervention. Data on functional performance, relative weight and body fat distribution, quality of life, and cognitive function will be recorded by the multidisciplinary group during the first and last weeks of the exercise intervention. Heart rate variability, exercise adherence, feasibility, and safety will be recorded daily during the intervention by the physical exercise graduate.

#### 2.4.1. VO_2peak_

In accordance with the Spanish Cardiology Society, a modified version of the Bruce protocol will be used as the treadmill test, as it is indicated for elderly people and those considered high risk. Unlike the Bruce protocol, in the modified version, stage 1 starts at a 0% gradient, walking at 45.5 m/min. During stages 2 and 3 (at 3′ and 6′), the percentage gradient increases by 5% each time. Therefore, 6′ more exercise is performed at low intensity. From there, the percentage gradient will increase by 2% every 3′, and the walking speeds will be 67 m/min, 91 m/min, 112.5 m/min, and 136.8 m/min in Stages 4–7, respectively [40]. Pulmonary gas exchange will be measured from the beginning of the rest period until the end of the test using a metabolic system (Parvo TrueOne 2400, ParvoMedics Inc., East Sandy, UT, USA or MetaMax3B, Cortex Biophysik, GmbH, Leipzig, Germany). Prior to the test, participants will be asked to avoid exercise, caffeine, and tobacco for 24 h and food for 2 h but to take all the usual medications. In accordance with Taylor et al. (2017) [41], VO_2_peak will be determined from the average of the two highest values attained in 10 s periods, disregarding outliers when the difference between the two highest VO2 values is greater than 200 mL/min divided by the participant’s body weight. Additionally, an electrocardiogram recording will be taken during the rest period and during the test. Endothelial parameters, such as systolic and diastolic blood pressure and heart rate, will be recorded during the rest period and during the 1 min and 5 min post tests. Work parameters, such as the protocol stages performed, the maximum velocity and gradient reached, the test duration, and the reason for ending it will also be recorded.

#### 2.4.2. Heart Rate Variability

Taking into account the sample characteristics, we propose that the HRV be measured over a 60 s period with the subject lying down at rest just prior to the training session. This procedure, modified from Plews et al. (2017) [42], will allow the multidisciplinary team to manage data recording, maintaining the same conditions and position, thus obtaining more reliable data [43]. The HRV4Training smartphone application will be used. This tool was validated by Plews et al. (2017) [42] and employed in several studies, as it facilitates data measurements, providing either temporal (rMSSD, pNN50, and SDNN) or frequential (LF and HF) measurements using photoplethysmography. Data can be downloaded directly to a computer in the form of an Excel sheet.

#### 2.4.3. Functional Performance

To measure functional capacity, the battery of tests included in the Senior Fitness Test [44] will be used, as these have been employed in previous studies on older people in which a program based on high-intensity training was implemented [9]. The following variables will be assessed: (a) static equilibrium by means of the monopodal equilibrium test; (b) the strength of the upper body with a bicep curl test; (c) the strength of the lower train using a test involving getting up from a chair and sitting; (d) agility with the timed-up and go (TUG) test; (e) gait speed, involving a test walking for 10 m; and (f) cardiorespiratory fitness with the 6 min walk test.

#### 2.4.4. Relative Weight and Body Fat Distribution Measures

Height will be measured using a measuring rod (Seca 213), and the body mass index (BMI) will be calculated according to the following formula: BMI = kg/m^2^. As body fat distribution can vary substantially and considering that abdominal adiposity has been associated with coronary heart disease [45,46], the waist-to-hip ratio will be measured according to the International Society for the Advancement of Kinanthropometry Standards [47]. The waist-to-height ratio will also be measured because recent systematic reviews with meta-analyses proposed this measure as a predictor of cardiovascular risk factors [48].

#### 2.4.5. Quality of Life

The MacNew QLMI post-myocardial infarction questionnaire will be completed. The questionnaire includes 27 specific items assessing the quality of life of patients who have suffered a myocardial infarction. The validated Spanish version of the MacNew QLMI will be used, which has demonstrated good reproducibility and internal consistency [49], preserving the “emotional”, “physical”, and “social” dimensions in its last modification [50].

#### 2.4.6. Blood Analysis

Following the Taylor et al. (2017) [41] protocol, blood analysis will include the lipid profile, detection and quantification of plasma insulin and its resistance, assessment of C-reactive protein, and inflammatory adipokines (leptin, adiponectin, tumor necrosis factor-a, interleukin-6, and appetite hormones).

#### 2.4.7. Cognitive Function

To determine brain function, several test will be used: (i) the Almeria spatial memory recognition test (ASMRT), a simple virtual-reality-based task for specific populations, such as older adults, that has a proven sensitivity to gender differences [51]; (ii) the virtual version of the Walking Corsi test (VR_WalCT) to measure topographical memory, given that [52] its equivalency to the real environment version has been proven; and (iii) the trail-making test (TMT) described by Bowie and Harvey (2006) [53] to detect neurological disease and neuropsychological impairment through the cognitive domains of processing speed, sequencing, mental flexibility, and visual motor skills.

#### 2.4.8. Exercise Adherence, Feasibility, and Safety

Adherence will be assessed by the number of sessions completed by the participants. At least 80% of sessions should be completed to consider adherence successful. Feasibility will be considered compliance with the exercise protocol and will be recorded as the percentage of participants that finish every training session. Safety will be measured as the number of adverse events occurring during the training sessions. In accordance with Villelabeitia-Jaureguizar et al. (2017) [54], these will be recorded as mild, moderate, or severe. Moreover, their relation to the exercise session will also be taken into account. All these variables will be recorded by the multidisciplinary group present at the exercise sessions.

The time schedule for enrolment, interventions, and assessments is shown in Figure 2.

### 2.5. Sample Size and Power

Based on the mean standard deviation established for the VO_2peak_ in a previous study by Moholdt et al. (2009) [55] (SD = 4.75 mL/kg/min) and an estimated error (*d*) of 1.7, a valid sample size providing a 95% confidence interval (CI) comprise 30 patients in each group (*n* = *CI*^2^ × *d*^2^/*SD*^2^). Therefore, a final sample size of 30 for each group (a total sample size of 60 patients) will provide a power of 97% if between and within a variance of 1. Calculations to establish the sample size will be performed using RStudio 3.15.0 software (RStudio, Boston, MA, USA). The significance level will be set at *p* ≤ 0.05.

### 2.6. Recruitment

The sample will be recruited from patients admitted to the rehabilitation section of the hospital following cardiological screening. A nurse will contact them by telephone to invite them to an informative talk. During this meeting, patients will be informed about the importance of a healthy lifestyle, maintaining physical and mental health, and being active, as well as about the characteristics of the rehabilitation program itself and the tests that will be implemented. Patients will also be encouraged to attend the pre-test and post-test sessions and to attend at least 90% of the training sessions. Information will be provided by the rehab doctor, the physiotherapist, the nurse, and the physical exercise graduate. If the patients agree to participate, they then have to sign the written consent form.

### 2.7. Allocation and Blinding

A block randomization method will be used to allocate participants to the groups, which will contain equal sample sizes. Blocks will be chosen randomly by tossing a coin to determine participant assignment to the groups. This procedure will be carried out by the data monitoring committee. The participants and the data monitoring committee will be blinded to the group assignment exercise.

### 2.8. Data Analysis

The data monitoring committee will have full access to the complete final dataset. The data will be analyzed using Jamovi (Jamovi Project 2018, version 0.9.1.7, Jamovi, Sydney, Australia) and RStudio 3.15.0 software. Prior to data analysis, the Kolmogorov–Smirnov test and the Levene test will be performed to determine the normal distribution of the variables and the homogeneity of variance. For a variable to be considered as having a normal distribution, 95% of values will have to be within two standard deviations of the mean. Descriptive data will be reported as mean ± SD and range. All data will be analyzed based on the intention-to-treat principle (the last observation carried forward). If the sample is normally distributed, the Student’s *t*-test for independent samples will be used to compare the two intervention groups for the pretest, whereas ANCOVA will be used for the post-test, using the basal scores and age as covariables. Moreover, the Student’s *t*-test for related samples will be calculated to compare variables before and after the intervention in each group. If the sample is non-parametric, Kruskal–Wallis and the Mann–Whitney U tests will be used, respectively. The standardized mean differences (Cohen’s effect size) will be calculated together with the 95% confidence intervals [56]. The effect sizes (ESs) will be calculated using Cohen’s d (Hopkins). The relationship between variables will be assessed using the Pearson r correlation coefficient. If r is higher than 0.7, the determination coefficient (r^2^) will be used to determine the percentage of Y variation with regard to the X variation. Significance will be accepted at *p* ≤ 0.05.

### 2.9. Monitoring

A data monitoring committee will be set up before commencing the protocol, comprising members of the University of Almería. Statistical analyses will be carried out in strict confidence. Based on the data monitoring committee’s advice, the trial’s multidisciplinary group members, together with the cardiologist, will decide whether or not to modify the trial intake.

An adverse event will be considered any untoward medical occurrence involving a subject, regardless of the possibility of a causal relationship. Adverse events will be registered from the moment the subject has signed the written consent. If the adverse event occurs after the subject has signed the informed consent but has not yet started the intervention, it will be considered as not related to the study’s exercise program. If the adverse event occurs after the subject has finished the intervention, it will likewise not be considered to be caused by the study protocol procedure. For this study, sickness, loss of consciousness, weakness, chest pain, or other acute pain will be considered serious adverse events.

### 2.10. Ethics and Dissemination

The protocol, the informed consent template contained in Appendix A, and other requested documents (if any) will be reviewed and approved by the Bioethical Committee of the University of Almería and the Bioethical Committee of University Hospital Torrecárdenas. Following the initial review and approval, this protocol will be reviewed by the researchers at least once a year at Clinicaltrials.org, where it is registered with ID NCT04150952. The last log was recorded on 07/20/2022 (protocol amendment number 03). Both bioethical committees will be informed of any protocol modifications that might have an impact on the study. Minor corrections that have no effect on the way the study is conducted will be agreed upon by the researchers and documented in a memorandum. All study-related information will be stored securely at the study site. All records that contain names or other personal identifiers, such as locator forms and informed consent forms, will be stored separately from the study records and identified by a code number. Forms, lists, logbooks, appointment books, and any other listings that link participant ID numbers to other identifying information will be stored in a separate, locked file.

## 3. Conclusions

This study protocol for randomized controlled trials offers a recommendation on the therapeutic approach for patients undergoing cardiac rehabilitation, taking into account the results obtained in the experimental studies and systematic reviews with metaanalisis carried out so far. For this purpose, the heart rate variability is proposed as a key variable to guide the training diary according to the patient’s parasympathetic nervous system activation. This should enhance the training individualization thus increasing it’s efectiveness. Therefore, after our protocol implementation it is expected to find better functional improvements in the HRV-based training group than in the HIIT-based training group, providing a new and simple method to improve the training control in patients after stroke.

## Figures and Tables

**Figure 1 ijerph-19-10868-f001:**
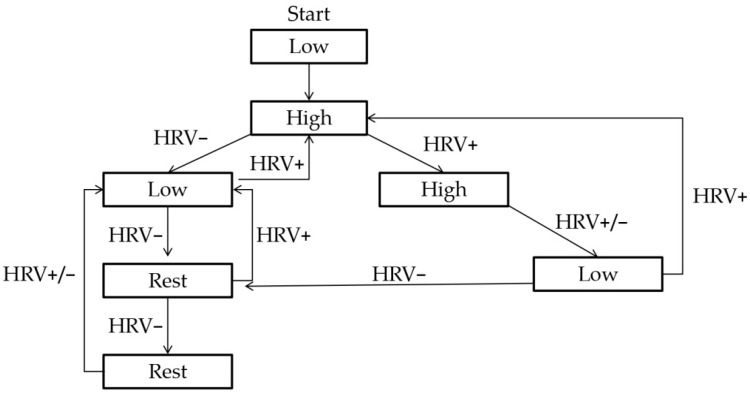
HRV-guided training schema [36]. When LnrMSSD7-d is within the normal range, high-intensity training sessions will be prescribed. If LnrMSSD7-d falls below the normal range, low intensity or rest will be prescribed. Low = exercise at 65% of maximal heart rate, High = exercise at 85% more of maximal heart rate, Rest = resting day, HRV+ = increased or unchanged HRV; HRV− = decreased HRV.

**Figure 2 ijerph-19-10868-f002:**
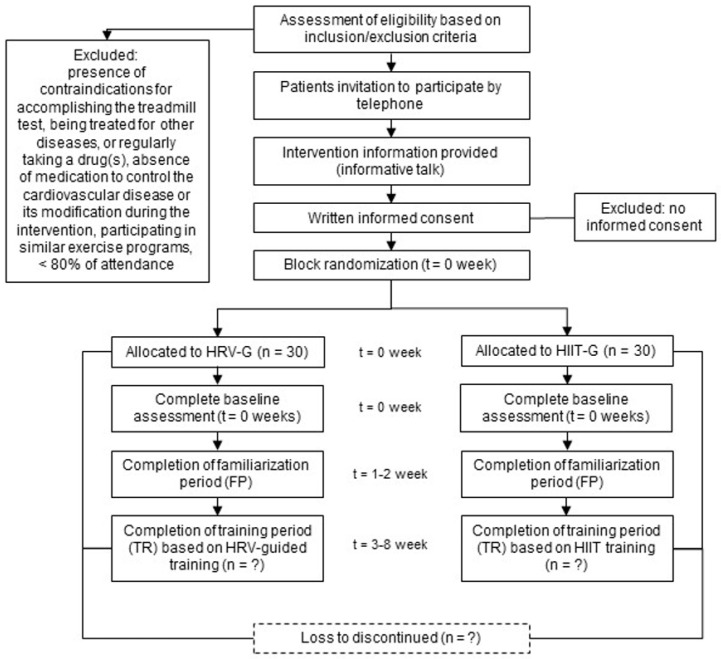
Schedule of enrolment, interventions, and assessment. HRV-G = heart-rate-variability-based training group; HIIT-G = HIIT-based training group.

**Table 1 ijerph-19-10868-t001:** Periodization and training distribution for HIIT-G.

**Period**	**Week**	**Day/Date**	**Session**	**Time**		**Intensity**		**Continuous Training**
FP	1	Day 1	S1	25′–30′		65–70% max HR RPE 6–7		
		Day 2	S2					
		Day 3	S3					
	2	Day 4	S4	30′–35′		70–75% max HR RPE 6–7		
		Day 5	S5					
		Day 6	S6					
**Period**	**Week**	**Day/Date**	**Session**	**Time**		**Intensity**		**Intervals**
				**W**	**Rest**	**W**	**Rest**	
TP	3	Day 7	S7	1′30″	1′30″	85% max HR RPE 8–10	60–70% max HR RPE 5–6	8
		Day 8	S8					8
		Day 9	S9					8
	4	Day 10	S10	1′30″	1′30″	85% max HR RPE 8–10	60–70% max HR RPE 5–6	8
		Day 11	S11					8
		Day 12	S12					8
	5	Day 13	S13	1′30″	1′15″	90–95% max HR RPE 9–10	60–70% max HR RPE 5–6	8
		Day 14	S14					8
		Day 15	S15					8
	6	Day 16	S16	1′30″	1′15″	90–95% max HR RPE 9–10	60–70% max HR RPE 5–6	9
		Day 17	S17					9
		Day 18	S18					9
	7	Day 19	S19	1′30″	1′	95–100% max HR RPE 9–10	60–70% max HR RPE 5–6	9
		Day 20	S20					9
		Day 21	S21					9
	8	Day 22	S22	1′30″	1′	95–100% max HR RPE 9–10	60–70% max HR RPE 5–6	9
		Day 23	S23					9
		Day 24	S24					9

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
