# Peer review of "HRV-Guided Training for Elders after Stroke: A Protocol for a Cluster-Randomized Controlled Trial"

_ijerph, 2022, doi:10.3390/ijerph191710868_

Round 1
Reviewer 1 Report
Thank you for the opportunity to review the manuscript entitled “HRV-Guided Training for Olders after Stroke: A Protocol for a Cluster-Randomized Controlled Trial”
The manuscript is well written and contains important information for reproducing the protocol in question, despite the authors commenting that the SPIRIT guidelines were used. I suggest the use of the SPIRIT checklist for better elaboration of the protocol. I describe below the missing items for the guideline (https://www.spirit-statement.org/)
3. Protocol version: Date and version identifier
Interventions
11b Criteria for discontinuing or modifying allocated interventions for a given trial participant (eg, drug dose change in response to harms, participant request, or improving/worsening disease)
11c Strategies to improve adherence to intervention protocols, and any procedures for monitoring adherence (eg, drug tablet return, laboratory tests)
Harms
22 Plans for collecting, assessing, reporting, and managing solicited and spontaneously reported adverse events and other unintended effects of trial interventions or trial conduct
Access to data
29 Statement of who will have access to the final trial dataset, and disclosure of contractual agreements that limit such access for investigators
I suggest the reference:
Raimundo RD, de Abreu LC, Adami F, Vanderlei FM, de Carvalho TD, Moreno IL, Pereira VX, Valenti VE, Sato MA. Heart rate variability in stroke patients submitted to an acute bout of aerobic exercise. Transl Stroke Res. 2013 Oct;4(5):488-99. doi: 10.1007/s12975-013-0263-4. Epub 2013 May 22. PMID: 24323375.
Reviewer 2 Report
Reviewers' comments to Authors:
The study addresses an important issue that HRV-guided training can be used in cardiac rehabilitation in older post-stroke patients. In the present study, the authors hypothesized that HITT training and HRV-guided training can improve patient outcomes.
The Authors should consider the following comments:
- This is very interesting study. The introduction is well written, presents the problem. Authors should add that post-stroke patients are characterized by autonomic imbalance which is common in many chronic and autoimmune diseases. It is noteworthy that post-stroke patients are characterized by hypertension, which is also associated with dysautonomia. Chronically overactive sympathetic nervous system increases mortality in these groups. Moreover, ANS dysfunctions should be considered at each stage of the diagnostic and treatment processes, as a predictor for the patient's clinical condition. Cardiac and autonomic dysfunction should be considered in pharmacological approach and rehabilitation approach. Please add some references.
- Methods: Please add more information about inclusion/exclusion criteria in study protocol (Page 4)
- Authors mentioned that HRV4Trianing smartphone application will be used. Please add advantages and disadvantages of this device, as compared to standard methods. Does this HRV method asses LF/HF ratio?
- Which parameters of HRV were included into analysis? (in which domain time/frequency)
- Have the patients been examined by cardiologists/neurologist?
- It is a well-designed study, with clear methodology
